# Innovative Controlled-Release Systems for Fucoxanthin: Research Progress and Applications

**DOI:** 10.3390/pharmaceutics17070889

**Published:** 2025-07-08

**Authors:** Shiyan Wang, Mengran Guo, Zhaohui Jin

**Affiliations:** Department of Pharmacy, West China Hospital, Sichuan University, Chengdu 610041, China

**Keywords:** fucoxanthin, poor aqueous solubility, controlled-release system, extracellular vesicles, bioavailability

## Abstract

Fucoxanthin, a marine-derived carotenoid primarily sourced from algae and microalgae, holds significant potential for pharmaceutical and nutraceutical applications. However, its highly unsaturated structure presents critical challenges, including structural instability, poor aqueous solubility, and limited bioavailability. These restrict its application despite its abundant natural availability. Recently, various controlled-release nanotechnologies have been applied to improve the properties of fucoxanthin formulations. In this review, we systematically summarized the bioactivities of fucoxanthin and highlighted recent advancements in controlled-release systems designed to address the limitations. These controlled-release systems mainly use natural or synthetic organic materials and are employed to develop various formulations, including emulsions, nanoparticles, nanofibers, and nanostructured lipid carriers. In addition, the emerging bioinspired drug delivery systems, particularly extracellular vesicles and cell-membrane-derived biomimetic systems, have gained prominence for their immunocompatibility and ability to penetrate physiological barriers, which is regarded as superior encapsulation vesicles for fucoxanthin. Focusing on innovations, we discussed the state-of-the-art delivery systems for fucoxanthin encapsulation and emphasized their roles in improving biosafety, enhancing bioavailability, preserving bioactivity, and optimizing therapeutic performance across various disease models. These insights will provide promising guidance for engineering controlled-release platforms and will aim to unlock fucoxanthin’s full potential in drug development and dietary supplement formulations.

## 1. Introduction

Fucoxanthin is an algae-specific carotenoid predominantly derived from brown seaweeds and diatoms, accounting for over 10% of total carotenoid production in marine ecosystems [1,2,3]. Notably, this compound not only possesses fundamental antioxidant activity but also demonstrates multidimensional health-promoting properties, including antimicrobial effects and anti-inflammatory characteristics. These attributes enable its efficacy in preventing chronic diseases such as diabetes, obesity, cancer, inflammatory disorders, cardiovascular diseases, and neurodegenerative pathologies, thereby establishing its pivotal role in nutraceutical development and therapeutic applications [4,5,6,7,8,9,10,11,12,13,14,15,16]. These attributes position it as a critical compound in nutraceutical and pharmaceutical research [17,18].

Nevertheless, inherent molecular constraints (Figure 1) limit the applications of fucoxanthin. The 5,6-epoxy and allenic bond structures within its chemical architecture are known to be the cause of the substance’s poor chemical stability, rendering it highly susceptible to photodegradation, thermal decomposition, and oxidative stress [19,20]. Consequently, fucoxanthin-based formulations require stringent processing, storage, and transportation protocols, significantly restricting their application in the medical and food industry [21,22,23]. Its lipophilic nature and low aqueous solubility result in variable oral absorption, which is food-matrix-dependent [24]. Furthermore, fucoxanthin undergoes degradation in the acidic gastric environment and through pepsin-mediated enzymatic hydrolysis, culminating in compromised bioavailability and attenuated bioactivity post-oral administration [25].

To overcome these limitations, recent advancements have pivoted toward developing advanced controlled-release systems employing hydrophilic biomaterials. Empirical evidence confirms that such systems can significantly enhance stability, protect against gastrointestinal tract (GIT)-induced degradation, and improve absorption efficiency [26,27]. The focus of contemporary research in this field is predominantly oriented towards the development of innovative encapsulation technologies and biocompatible materials. This involves the design of delivery systems, such as nanoemulsions, nanoparticles, and extracellular vesicles, combined with organic carriers composed of protein–polysaccharide complexes [28,29]. Critically, physical encapsulation methodologies not only amplify the photothermal stability of active compounds by forming protective barriers but also leverage thermally insulating networks to modulate release kinetics under extreme conditions.

This review synthesizes recent advancements in fucoxanthin delivery systems, analyzing their compositional design, fabrication methodologies, and functional outcomes. An emphasis is placed on the system-specific impacts on stability, release, bioavailability enhancement, bioactivity retention, and therapeutic performance in disease models.

## 2. The Biological Activity of Fucoxanthin

Fucoxanthin has been demonstrated to exhibit foundational biological activities, including antioxidant, anti-inflammatory, antibacterial, and anticancer properties [30,31,32,33,34]. Recent studies have further elucidated its novel functionalities in anti-obesity, anti-aging, and neuroprotective effects, significantly expanding its therapeutic and nutraceutical application potential. Figure 2 demonstrates the multi-target bioactivities of fucoxanthin.

### 2.1. Antioxidant Activity

The exceptional antioxidant capacity of fucoxanthin stems from its unique epoxy groups and allene bonds, which facilitate the efficient scavenging of oxygen free radicals. Studies demonstrate its protective effects across multiple organ oxidative stress injury models: in renal damage induced by cadmium, carbon tetrachloride (CCl4) exposure, and ischemia–reperfusion, fucoxanthin effectively reduces serum creatinine and blood urea nitrogen levels while enhancing renal antioxidant capacity [35,36,37]. Through the inhibition of endoplasmic reticulum stress-mediated apoptosis pathways, this molecule alleviates intervertebral disc degeneration in rat models [12]. In osteoarthritis models, it exerts dual chondroprotective and antioxidant effects by suppressing the expression of matrix metalloproteinase (MMP) family members and interleukin-1β (IL-1β), thereby blocking the phosphorylation processes of p65, JNK, and p38 proteins [38]. Furthermore, fucoxanthin markedly mitigates oxidative stress-induced DNA damage in cardiomyocytes, inhibiting both apoptosis and pathological calcification processes [39].

### 2.2. Anti-Inflammatory Activity

Fucoxanthin’s potent anti-inflammatory effects underpin its utility in therapeutic regimens and preventive healthcare strategies. The compound effectively suppresses abnormal upregulation of IL-1β, TNF-α, iNOS, and COX-2 in high-fat diet-induced obese murine models while attenuating macrophage-mediated inflammatory responses [40]. In cisplatin (CP)-induced testicular macrophage injury, fucoxanthin reduces reactive oxygen species (ROS) and malondialdehyde (MDA) levels, concurrently stabilizing mitochondrial membrane potential to exert cytoprotective effects. Administration of fucoxanthin significantly elevates testosterone concentrations and α-glucosidase activity, enhances sperm count with improved morphological integrity, and restores seminiferous tubule architecture [41]. Within lipopolysaccharide (LPS)-induced septic models, it potently inhibits the expression of pro-inflammatory cytokines including IL-6, IL-1β, and TNF-α. Mechanistic studies confirm its capacity to suppress LPS-induced phosphorylation of the NF-κB signaling pathway and inhibit nuclear translocation of NF-κB at the cellular level [42]. Furthermore, in hepatic fibrosis inflammation models, fucoxanthin not only mediates therapeutic effects through these anti-inflammatory mechanisms but also reduces leukocyte infiltration in injured liver tissue [43].

### 2.3. Antibacterial Activity

Through distinct bactericidal mechanisms, fucoxanthin offers broad-spectrum antimicrobial activity against Gram-positive and Gram-negative pathogens. As a photosensitizer, it demonstrates potent antibacterial activity against *Staphylococcus aureus* via photodynamic mechanisms: polyunsaturated fatty acids in its structure undergo photooxidation to generate singlet oxygen (^1^O_2_) and reactive oxygen species (ROS), inducing bacterial lipid peroxidation and cell death, thereby synergistically enhancing cefotaxime’s antimicrobial efficacy [33,44]. The compound also inhibits critical metabolic enzymes, including α-amylase and trypsin, exhibiting broad-spectrum suppression against both Gram-positive (*Listeria monocytogenes*, *S. aureus*) and Gram-negative (*Salmonella enterica*) pathogens in intestinal environments, with a minimum inhibitory concentration (MIC) of 0.2 μg/mL, indicating therapeutic potential for gastrointestinal infections [45]. Notably, fucoxanthin achieves superior antitubercular effects by targeting mycobacterial cell wall biosynthesis enzymes—UDP-galactopyranose mutase (UGM) and arylamine-N-acetyltransferase (TBNAT)—demonstrating lower half-maximal inhibitory concentration (IC50) and higher inhibition efficiency compared to isoniazid [46]. However, current evidence confirms its limited activity against strict anaerobes [47].

### 2.4. Anticancer Activity

Significant antitumor efficacy positions fucoxanthin as a promising candidate for cancer therapeutics. Research indicates this compound exerts anticancer effects through multiple mechanisms: suppressing cancer cell proliferation, angiogenesis, and metastatic invasion; modulating tumor-associated miRNA expression; and inducing programmed cell death pathways, including ferroptosis, cell cycle arrest, apoptosis, and autophagy [48,49]. For instance, in gastric cancer MGC-803 cell models, fucoxanthin triggers apoptosis and induces G2/M phase cell cycle arrest [50]. In breast cancer HLEC cells, it significantly downregulates VEGF-C, VEGFR-3, and NF-κB expression levels while inhibiting phosphorylation activation of Akt and PI3K [51]. Existing studies have confirmed its therapeutic efficacy against various malignancies such as tongue carcinoma, prostate cancer, lung cancer, and bladder cancer, underscoring its substantial pharmaceutical development value [49,52].

### 2.5. Anti-Obesity Activity

Fucoxanthin demonstrates remarkable anti-obesity activity, with therapeutic efficacy against obesity-related complications. Studies reveal its direct modulation of adipocyte biology through the inhibition of glucose uptake in mature adipocytes and suppression of mid- and late-phase adipocyte differentiation [53,54]. The compound exerts its effects by targeting lipid-metabolizing enzymes—upregulating key fatty acid oxidation enzymes such as carnitine palmitoyltransferase-1 (CPT1) and cholesterol 7α-hydroxylase 1 (CYP7A1) while inhibiting cholesterol regulatory enzymes, including 3-hydroxy-3-methyl-glutaryl coenzyme A (HMG-CoA) and acyl-CoA cholesterol acyltransferase (ACAT) [55,56,57]. Furthermore, fucoxanthin modulates gut microbiota composition to potentiate its anti-obesity effects [58,59,60,61,62,63]. Clinical validation is implied by meta-analysis, demonstrating that fucoxanthin-enriched seaweed supplementation significantly reduces body mass index (BMI), fat mass, total cholesterol, and low-density lipoprotein cholesterol (LDL-C) levels in human subjects [64].

### 2.6. Anti-Aging Efficacy

The potential anti-aging efficacy of fucoxanthin is well supported by experimental evidence [65]. As demonstrated in the relevant studies, fucoxanthin has been shown to significantly extend the median lifespan of *Drosophila melanogaster* (15.7% increase) while concomitantly enhancing fecundity, fertility, intestinal barrier integrity, and circadian sleep rhythms [66]. Concurrently, in models of photoaging, fucoxanthin has been observed to suppress the UVA-induced pro-inflammatory cytokines TNF-α and IL-6 in the skin of BALB/c mice, whilst concomitantly upregulating anti-inflammatory IL-10 production. This has been shown to reduce post-radiation wrinkle formation and desquamation, accelerate re-epithelialization, and enhance antimicrobial activity against *Cutibacterium acnes* [67,68]. Mechanistically, fucoxanthin protects fibroblasts from senescence by activating the Nrf2/ARE pathway, modulating ribosome biogenesis, lipid metabolism, and cell cycle regulation. These effects involve key signaling pathways associated with the aging process, including Wnt, JAK-STAT, and FoxO [69]. The co-administration of low-molecular-weight fucoidan (LMWF) with fucoxanthin demonstrates synergistic cardioprotective effects, evidenced by the coordinated regulation of cardiac senescence markers in aged mice through the concurrent modulation of tricarboxylic acid (TCA) cycle flux, glycolytic activity, and steroid hormone biosynthesis pathways (*p* < 0.05). This, in turn, has been shown to result in improvements in ventricular rhythm and skeletal muscle function [70]. Furthermore, in vivo studies have confirmed the ability of fucoxanthin to mitigate oxidative stress-induced premature senescence in retinal cells [71].

### 2.7. Neuroprotective Properties

Preclinically, fucoxanthin demonstrates significant neuroprotective properties [72,73,74]. Mechanistic studies reveal its capacity to alleviate neuropathic pain by downregulating transient receptor potential vanilloid 1 (TRPV1) expression in trigeminal ganglion neurons [75]. In valproic acid (VPA)-induced rat autism models, fucoxanthin exhibits dose-dependent enhancement of spatial memory, attenuation of hyperalgesia, and improvements in social interaction, locomotor activity, balance coordination, and motor function. Concurrent neuromodulatory effects include elevated γ-aminobutyric acid (GABA) levels with reduced glutamate concentrations in cortical and cerebellar regions [76]. Furthermore, fucoxanthin inhibits β-amyloid (Aβ) fibril and oligomer formation, significantly mitigating cognitive deficits in Aβ oligomer-injected Alzheimer’s disease murine models. The therapeutic mechanisms involve oxidative stress suppression through 2.1-fold increased superoxide dismutase (SOD) activity and 60% reduction in malondialdehyde (MDA) levels (*p* < 0.01), accompanied by a 4-fold upregulation of brain-derived neurotrophic factor (BDNF) expression in hippocampal tissue, along with a 3-fold enhanced choline acetyltransferase (ChAT)-positive neuronal density in the CA1 region compared to untreated controls [77].

### 2.8. Applications of Encapsulated Fucoxanthin in Various Disease Models

In recent years, encapsulated fucoxanthin has demonstrated therapeutic potential in obesity, non-alcoholic fatty liver disease (NAFLD), antioxidant, colitis, and tumor (cell/murine) disease models. Oliyaei et al. developed a fucoidan–*N. sativa* oil composite emulsion-loaded fucoxanthin that exhibited anti-obesity activity in obese rat models by reducing weight-related parameters (BMI, Lee index), liver weight, lipid accumulation, and hepatic injury. However, the oil-carried fucoxanthin showed limited efficacy in reducing fasting blood glucose (>120 mg/dL vs. 79.50 ± 10.03 mg/dL for free fucoxanthin) while demonstrating an enhanced hepatic ALP reduction (203.83 ± 13.26 IU/L vs. 235.50 ± 12.82 IU/L for free fucoxanthin) [78]. Li et al. employed hydroxypropyl-β-cyclodextrin (HP-β-CD) fibers to encapsulate fucoxanthin, confirming its capacity to ameliorate metabolic disorders in obese mice and alleviate high-fat diet-induced testicular damage. Notably, mice treated with fucoxanthin/HP-β-CD nanofibers achieved 14% greater weight loss compared to free fucoxanthin-treated counterparts [79]. Wu et al. established a glycyrrhetinic-acid-modified extracellular vesicle (GA-EVs) delivery system that effectively delayed NAFLD progression through lipid synthesis regulation, fat deposition reduction, and lipid metabolism disorder amelioration. This system achieved a 2-fold hepatic accumulation enhancement, confirming liver-targeting superiority [80]. Wang et al. constructed a kelp-derived nanocellulose/casein nanodelivery system that alleviated oxidative stress in HepG2 cells via Nrf2/HO-1/NQO1 signaling pathway activation, inhibited free fatty acid (FFA)-induced lipid droplet formation, and enhanced hepatic targeting through prolonged intestinal retention. Compared to free fucoxanthin, this carrier system demonstrated a superior retention stability (56.12 ± 1.89% vs. <25% for free fucoxanthin after 14 days) and 10% increased antioxidant capacity [81]. Liang et al. achieved colon-targeted anti-inflammatory effects using extracellular vesicle-encapsulated fucoxanthin, involving mechanisms of H_2_O_2_-mediated oxidative damage scavenging, macrophage anti-inflammatory polarization induction, pro-inflammatory cytokine secretion suppression, colonic histopathological injury repair, and gut microbiota modulation to elevate short-chain fatty acid levels. The encapsulated form exhibited a 2.5-fold higher gastric retention (58.09% vs. 23% for free fucoxanthin) and 20% increased intestinal retention compared to free fucoxanthin, with iNOS activity reduced from 4.9 U/mg prot to 1.2 U/mg prot in LPS-induced inflammation models [82]. Wu et al. developed hydroxyethyl starch-cholesterol self-assembled nanoparticles co-delivering fucoxanthin and Twist siRNA, demonstrating synergistic antitumor efficacy in triple-negative breast cancer (TNBC) through tumor cell proliferation/metastasis inhibition, apoptosis promotion, aberrant angiogenesis blockade, and cancer-associated fibroblast (CAF) activation/collagen deposition suppression in tumor microenvironments, while enhancing drug penetration into tumor stroma. This combinatorial strategy reduced post-treatment cell confluence from 54.23 ± 2.89% (free fucoxanthin) to 22.36 ± 2.65% in migration assays and achieved 33% greater tumor volume reduction in 4T1 xenograft models compared to the free fucoxanthin-treated group [83]. Wang et al. designed a flaxseed gum–whey protein composite system that augmented cellular apoptosis via ROS overload and mitochondrial dysfunction, with the potential inhibition of the MAPK-PI3K pathway to suppress orthotopic liver tumor growth in nude mice xenografts. These therapeutic effects exhibited dose-dependent responses [84].


## 3. Novel Controlled-Release Systems for Fucoxanthin Delivery

To further enhance fucoxanthin bioavailability and maximize its therapeutic efficacy, recent advancements have focused on developing novel controlled-release delivery systems using biocompatible materials. This section critically reviews innovative fucoxanthin-loaded delivery platforms, including nanoemulsions, nanoparticles, lipid-based nanocarriers, nanofibers, and biomimetic extracellular matrix constructs. Table 1 systematically delineates the core parameters of fucoxanthin delivery systems over the past three years, encompassing structural composition, fabrication methodologies, experimental models, particle size distribution, and principal pharmacodynamic outcomes. Figure 3 illustrates various controlled-release systems.

### 3.1. Emulsions

Emulsions, as common controlled-release systems, constitute biphasic systems comprising two or more immiscible liquid phases (oil and aqueous phases) stabilized thermodynamically by emulsifiers [85,86,87]. Based on interfacial hydrophilicity/hydrophobicity characteristics, emulsions are categorized into three fundamental types: conventional emulsions, nanoemulsions, and Pickering emulsions, with microstructures including oil-in-water (O/W), water-in-oil (W/O), and water-in-oil-in-water (W/O/W) configurations [88,89,90]. Notably, W/O/W emulsions enable the co-delivery of hydrophilic and hydrophobic bioactive compounds through multiphase compartmentalization [91,92]. For hydrophobic fucoxanthin delivery, O/W emulsions represent the standard encapsulation strategy. Innovatively, Wang et al. developed solid-in-oil-in-water (S/O/W) emulsions with programmable sequential release profiles via interfacial engineering, achieving the synergistic delivery of differentially hydrophobic actives (fucoxanthin and curcumin) [93]. Subsequent sections will detail composition optimization strategies (emulsifier selection, oil phase polarity) and process parameter modulation (high-pressure homogenization intensity, sonication amplitude) to enhance fucoxanthin photothermal stability. The recent three-year studies on fucoxanthin-loaded emulsion systems predominantly focused on O/W and S/O/W architectures.

**Table 1 pharmaceutics-17-00889-t001:** Summary of recent controlled-release systems for fucoxanthin.

Controlled-Release Systems	Carrier or Wall Materials	Emulsifiers	Fabrication Technologies	Experimental Category	Particle Size	Improvement	Reference
**Emulsions**							
O/W Emulsions	Fucoidan	*N. sativa* oil, Tween 80	High shear homogenization and ultrasound-assisted emulsification	In vitro and in vivo	181–184	(1) Increased encapsulation efficiency (89.94–91.68%). (2) Enhanced in vitro release rate (75.86–83.76%). (3) Reduced body/liver weight and alleviated hepatic lipid accumulation.	[78]
O/W Emulsions	Lysozyme	Phycocyanin	Stir	In vitro	31.06 ± 2.76 μm	(1) Optimized bioavailability during in vitro digestion. (2) Improved encapsulation stability within oil droplets	[94]
O/W Emulsion gels	Salmon by-product protein (SP), pectin	Corn oil	High shear homogenization	In vitro	3–15 μm	(1) Displayed protective release under simulated oral/gastric conditions and intestinal-targeted controlled release. (2) Improved fucoxanthin retention for 15 days at 25 °C. (3) Applicable to 3D printing-compatible food technologies.	[95]
S/O/W multilayer emulsions	Gliadin, carboxymethyl starch (CMS), propylene glycol alginate (PGA), carboxymethyl konjac glucomannan	Coconut oil	Anti-solvent precipitation method, rotary evaporate, ultrasound-assisted emulsification	In vitro and in vivo	24.8 ± 0.5 μm	(1) Stabilized against photothermal degradation. (2) Showed programmed sequential release profile: reduced gastric degradation and potentiated intestinal/colonic distribution.	[93]
**Polymer nanoparticles**							
Nanoparticles	Sodium alginate, chitosan	Sodium caseinate	Ionic gelation and polyelectrolyte complexation method	In vitro and in vivo	246.1 ± 7.9, 258.7 ± 9.4 nm	(1) Achieved controlled release and targeted delivery to intestinal epithelial cell membranes. (2) Amplified bioavailability in terms of absorption amount and residence time.	[96]
Nanoparticles	Chitosan, gelatin	/	Magnetic stir, ultrasound-assisted emulsification	In vitro	300 nm	(1) Potentiated encapsulation efficiency (83.88 ± 4.39%) and stability. (2) Upregulated cellular uptake and antioxidant activity.	[97]
Nanoparticles	Flaxseed gum	Whey protein	Anti-solvent precipitation method	In vitro and in vivo	348 ± 36 nm	(1) Promoted focal necrosis in tumor tissues. (2) Triggered tumor cell apoptosis in a dose-dependent way. (3) Caused anti-apoptotic factor (e.g., Bcl-2, CyclinD1, Ki-67) suppression and apoptotic (Bax) secretion, and inhibited tumor growth/metastasis.	[84]
Nanoparticles	Sodium alginate, chitosan	Tween 80	Ionic gelation method	In vitro and in vivo	227 ± 23 nm	(1) Potentiated antioxidant and anti-inflammatory activities. (2) Exhibited good cytotoxic effects against diverse cancer cell lines.	[98]
Nanoparticles	Oxidized paramylon	/	Anti-solvent precipitation method	In vitro	86.38–107.33 m	(1) Showed excellent storage stability and photostability. (2) Prevented premature release in gastric conditions. (3) Increased intestinal-phase release efficiency (72.17%). (4) Reduced ROS in insulin-resistant HepG2 cells. (5) Promoted cellular glucose uptake/utilization.	[99]
Nanoparticles	Hydroxyethyl starch, cholesterol	/	Rotary evaporated, ultrasound-assisted emulsification and magnetic stir	In vitro and in vivo	138.7 ± 0.9 mm	(1) Synergized drug penetration through tumor interstitial barriers for targeted delivery. (2) Reduced TNBC orthotopic tumor burden and suppressed pulmonary metastasis.	[83]
Nanoparticles	Kelp nanocellulose	Sodium caseinate	High shear homogenization, ultrasound-assisted emulsification	In vitro and in vivo	285.13 ± 5.85–309.3 ± 5.78 nm	(1) Encapsulation efficiency reached 82.2%, with >50% retention after 14-day storage. (2) Showed excellent cytocompatibility. (3) Enhanced cellular antioxidant enzyme activity and reduced ROS generation. (4) Improved intracellular delivery. (5) Suppressed excessive FFA-induced lipid droplet formation efficiency and bioavailability. (6) Boosted drug delivery efficiency and target-tissue accumulation. (7) Prevented premature release/degradation of active components in gastrointestinal tract.	[81]
Nanoparticles	Chitosan	Whey protein	Magnetic stir, freeze-drying technique and rotary evaporate	In vitro	171 ± 4 nm	(1) Upregulated water dispersibility and stability. (2) Improved encapsulation efficiency (≥93.6%).	[100]
Nanoparticles	Fucoidan	Pea protein isolated powder	PH-switchable molecular self-assembly method	In vitro	166.60 m	(1) Exhibited superior thermal and storage stability, effectively protecting fucoxanthin from degradation induced by pH fluctuations and high temperatures. (2) Demonstrated exceptional antioxidant capacity with significant radical scavenging activity.	[101]
Nanoparticles	Sodium alginate, κ-carrageenan, and Ca^2+^ crosslinking	/	Ionic gelation method and cross-linking method	In vitro	/	(1) Enhanced stability during in vitro digestion. (2) Maintained significant antioxidant activity with excellent thermal and photostability, indicating superior storage stability.	[102]
Nanoparticles	Glycosylated zein-based colloids	/	Maillard conjugate, PH-switchable molecular self-assembly method	In vitro and in vivo	<210.00 nm	(1) Enhanced thermal stability. (2) Improved digestive stability, micellization rate, and oral bioavailability.	[103]
Nanofibers	Hydroxypropyl-β-cyclodextrin	/	Magnetic stir, electrospinning method	In vitro and in vivo	499 ± 177 nm	(1) Improved aqueous solubility. (2) Enhanced thermal stability and aqueous dispersibility. (3) Anti-obesity and hypolipidemic effects: ameliorated body weight/dyslipidemia, alleviated hepatic steatosis and testicular injury	[79]
Nanofibers	Gelatin, zein	/	Coaxial electrospinning method	In vitro	662 ± 116 nm	(1) High-efficiency encapsulation (up to 98%). (2) Superior thermal stability and antioxidant activity. (3) Protection against harsh environmental conditions (temperature, light, UV radiation). (4) Enhanced biocompatibility: boosts antioxidant enzyme activity, inhibits ROS generation, maintains mitochondrial membrane potential	[104]
Nanostructured lipid carriers	Myristic acid (MA), palmitic acid (PA), stearic acid (SA), and arachidonic acid (AA)	Millard-modified zein	PH-switchable molecular self-assembly method	In vitro and in vivo	200.00~230.00 nm	(1) Improved water solubility. (2) Maximized encapsulation efficiency (>98.00%). (3) Enhanced bioaccessibility (increased from 43.00% to 60.00%). (4) Promoted oral absorption. (5) Significant upregulation of lipid transport-related protein expression.	[105]
Nanostructured lipid carriers	Coconut oil	Tween 80	High shear homogenization	In vitro and in vivo	248.98 ± 4.0 nm	(1) Exhibited outstanding stability. (2) Showed favorable in vitro release characteristics.	[106]
Nanostructured lipid carriers	Soy phosphatidylcholine, cholesterol	/	Ultrasonic film dispersio method	In vitro	98.28 nm	(1) Enhanced erythrocyte protective potential. (2) Inhibited hemolysis, photohemolysis, and heat-induced hemolysis.	[107]
**Biomimetic drug delivery system**							
Probiotics’ membrane vesicles	*Lactobacillus Plantarum*-derived extracellular vesicles	/	Centrifugated and sonication method	In vitro and in vivo	422 ± 9 nm	(1) Enhanced gastrointestinal stability. (2) Effective free radical scavenging. (3) Promoted macrophage M2 polarization. (4) Prevented colonic tissue damage and shortening. (5) Ameliorated colonic inflammatory responses. (6) Inhibited pro-inflammatory cytokines. (7) Modulated gut microbiota composition. (8) Increased short-chain fatty acid abundance in colon.	[82,108]
	*Lactobacillus paracasei*-derived extracellular vesicles	/	Centrifugated and sonication method	In vitro and in vivo	151–205 nm	(1) Displayed enhanced biocompatibility and improved stability. (2) Showed hepatic targeting efficiency. (3) Regulated lipid synthesis, reduced fat deposition, mitigated NAFLD progression, and ameliorated lipid metabolism disorders.	[80]
Bionic cell wall	2,2,6,6-tetramethyl-1-piperidinyloxy (TEMPO)-oxidized cellulose nanofiber (TCNF)	Lecithin from soybean, cholesterol	PH-switchable molecular self-assembly method, ultrasonic treatment	In vitro	239.33 nm	(1) Structural protection with improved environmental stability. (2) Achieved pH-responsive controlled release.	[109]

O/W emulsions, defined as biphasic systems consisting of an oil phase (O phase) dispersed within a continuous aqueous phase (W phase), have been shown to elevate the hydrophilicity of hydrophobic bioactive compounds. The categorization of emulsions is based on the dispersed-phase droplet size distribution, and the following classifications are made: conventional emulsions (100 nm–100 μm), nanoemulsions (10–100 nm), and microemulsions (2–50 nm) [110]. Research indicates that the relative impact of environmental factors on fucoxanthin stability in emulsions follows the order: pH > temperature > light exposure [111]. For instance, Bai et al. utilized phycocyanin (PC) and lysozyme (Lys) to prepare Pickering emulsions via a low-speed rotation method (300 rpm), yielding droplets with an average diameter of 31.06 ± 2.76 μm. At a PC:Lys mass ratio of 2:1, the nanocomposite exhibited a three-phase contact angle (θ) range of 58.9–84°, demonstrating optimal hydrophilicity and storage stability. The emulsion demonstrated a high level of efficacy in preventing the photodegradation and thermal degradation of fucoxanthin, achieving retention rates in excess of 90% after 40 min of UVB irradiation and over 95% after 10 min of heating at 60 °C [94]. Zhong et al. innovatively employed salmon byproduct protein–pectin complexes to fabricate fucoxanthin-loaded Pickering emulsion gels via high-speed shear homogenization (12,000 rpm, 2.5 min). Pectin enhanced interfacial adsorption energy, reducing oil droplet diameter to 3–15 μm and forming viscoelastic gels with thermal stability across 20–80 °C. Quantitatively, the system achieved an encapsulation efficiency of 95.63%, retaining 78.83 ± 0.04% fucoxanthin after 14 days at room temperature and 84.89% after 60 min at 90 °C. Functionally, its shear-thinning behavior, thixotropic recovery properties, and self-supporting strength enabled high-precision 3D food printing applications [95]. Furthermore, Oliyaei et al. developed fucoidan–*N. sativa* oil nanoemulsions via high-speed homogenization (12,000 rpm, 5 min) and ultrasonication (150 W, amplitude 80%, 10 min), achieving 181–184 nm average particle size and 89.94–91.68% encapsulation efficiency [78]. Critically, fucoidan stabilized emulsions through triple mechanisms: electrostatic repulsion, flocculation inhibition, and viscosity modulation [112].

The S/O/W emulsion comprises a triphasic system where the solid phase (S phase) is initially dispersed into the oil phase (O phase), followed by the secondary dispersion of the S/O co-dispersion into the aqueous phase (W phase). This “dual interfacial layer-triple phase” architecture enables synergistic co-encapsulation, controlled release, and protection of multiple actives. Distinct from conventional co-delivery systems, this multilayered structural system allows for the independent modulation of encapsulation loci and release kinetics for individual bioactive components [113]. Wang et al. engineered a triphasic emulsion using gliadin (Gli), carboxymethyl starch (CMS)/propylene glycol alginate (PGA) complex, and carboxymethyl konjac glucomannan (CMK). The fabrication protocol involved the following: (1) preparing fucoxanthin-Gli-CMK nanoparticles via antisolvent precipitation as the S phase; (2) dispersing the S phase into curcumin-loaded coconut oil (O phase) under ultrasonication; (3) incorporating the S/O dispersion into the W phase composed of CMS/PGA complex. The system achieved a fucoxanthin encapsulation efficiency of 96.0–96.1%. An increased PGA mass ratio enhanced the emulsion stability through the following: (a) thickening-induced oil droplet networking and (b) elevated W-phase viscosity to restrict droplet migration. The accelerated stability testing demonstrated no phase separation after 30-day storage at 25 °C, confirming long-term stability. Post 6 h UV irradiation, fucoxanthin retention remained 92.6–94.9% [93].

These findings substantiate that component optimization and structural innovations in emulsion systems markedly improve both the encapsulation efficiency and stability of fucoxanthin.

### 3.2. Polymer Nanoparticles

Biopolymer nanoparticles serve as delivery carriers for fucoxanthin, exhibiting advantageous properties, including biocompatibility, biodegradability, low toxicity, and non-antigenicity. The construction of particulate fucoxanthin delivery systems primarily involves three categories of biopolymers: proteins (e.g., zein and Gli), polysaccharides (e.g., starch and chitosan), and fatty acid derivatives (e.g., myristic acid and docosahexaenoic acid) [114,115,116]. Intermolecular forces, such as hydrophobic interactions, hydrogen bonding, disulfide bonds, and electrostatic interactions, play critical regulatory roles in the fabrication of fucoxanthin-loaded biopolymer nanoparticles [117,118,119,120]. While nanoparticles employing single polymers (e.g., polysaccharides, proteins, resins, or phospholipids) as wall materials can maintain fucoxanthin stability, the composite encapsulation of multiple polymer types confers synergistic advantages to the delivery system. Representative composite systems include protein–polysaccharide complexes, protein–protein complexes, polysaccharide–polysaccharide complexes, and polysaccharide–lipid complexes. These encapsulation systems can be classified into nanoparticles, nanostructured lipid carriers, and nano-tubules, based on morphological and compositional distinctions.

The development of protein–polysaccharide composite-based nanoparticle encapsulation technologies has garnered significant attention in recent years due to their superior biocompatibility, with chitosan–protein combinations (e.g., chitosan–whey protein) being widely employed [121]. Zhao et al. engineered an H-aggregate fucoxanthin nano-delivery system using chitosan/whey protein, achieving an encapsulation efficiency >93.6%. The system maintained excellent colloidal stability under 48 h storage at 4 °C, 25 °C, and 37 °C, while demonstrating remarkable tolerance to extreme pH conditions (pH = 1.0) [100]. In nanostructured lipid carrier (NLC) research, Kuang et al. developed a hydrophilic-stabilized carrier system via pH-driven self-assembly of Maillard-modified zein (MZ) with four dietary fatty acids: myristic acid (MA), palmitic acid (PA), stearic acid (SA), and arachidonic acid (AA). The resulting composite exhibited particle sizes of 200.00–230.00 nm, fucoxanthin encapsulation efficiency of 98.59–98.80%, and absolute zeta potential values ranging from +40.00 to +52.00 mV. These characteristics provided strong electrostatic repulsion to prevent particle aggregation in aqueous systems [105]. Tian et al. fabricated core–shell nanofibers via coaxial electrospinning, utilizing gelatin/zein (1:2 mass ratio) as the shell layer and fucoxanthin/zein as the core. The optimal parameters (core flow rate: 0.052 mL/h; shell–core flow ratio: 0.26:1) yielded the maximum encapsulation efficiency (98%) and loading capacity (10.32%). Thermogravimetric analysis revealed enhanced thermal stability, with the decomposition temperature increasing to 334.78 °C (free fucoxanthin: 299.79 °C). UV exposure testing showed an encapsulated fucoxanthin retention of 74.14% after 120 min exposure, marking a 109.8% improvement over free fucoxanthin (35.32%) [104].

### 3.3. Biomimetic Drug Deliver System

Biomimetic cellular components encompass structures such as biomimetic vesicles and biomimetic cell walls. Extracellular vesicles (EVs), natural nanoscale lipid bilayer membrane structures secreted by animal tissues and bodily fluids, exhibit exceptional immunocompatibility and pharmacokinetic properties [122]. Their ability to penetrate physiological barriers, evade phagocytic clearance mechanisms, and target pathological sites via the enhanced permeability and retention (EPR) effect positions them as ideal carriers for bioactive compound nano-delivery [123]. The lipid bilayer architecture further provides molecular interfaces for targeted ligand modification. However, the complex isolation processes and low yield of EVs significantly hinder their scalability in encapsulation systems. To address productivity limitations, researchers have developed stimulus-induced and synthetic EV production strategies. Mechanically extruded or ultrasonicated cell/bacterial membrane vesicles have been validated for efficient encapsulation [124]. For instance, Liang et al. synthesized membrane vesicles from *Lactobacillus* via ultrasonication, achieving a 2:1 fucoxanthin-to-vesicle ratio with a diameter of 422 ± 9 nm. The hydrophilic head groups of surface phospholipids enhanced aqueous dispersibility, while thermogravimetric analysis (TGA) demonstrated <10% fucoxanthin loss at 600 °C. The system effectively inhibited thermal/photodegradation, maintaining 88% fucoxanthin retention after 300 min at 80 °C (vs. 15% free form) and >80% retention post 150 min UV exposure (23% improvement over free fucoxanthin) [108]. Wu et al. optimized *Lactobacillus paracasei*-derived membrane vesicles for fucoxanthin encapsulation, reducing vesicle diameter to 205 nm while enhancing aqueous solubility and stability [80].

Plant cell walls, as natural mechanical support systems, utilize polysaccharide polymers (cellulose, hemicellulose, pectin) to maintain cellular morphological stability and confer environmental adaptation functions [125]. Inspired by this mechanism, polysaccharide-based biomimetic cell wall structures effectively protect liposomes from external perturbations. The research demonstrates that polysaccharide coating significantly enhances liposomal colloidal stability while reducing environmental sensitivity. As the core component of cell walls, cellulose emerges as an ideal natural polymer material due to its mechanical strength, biocompatibility, high specific surface area, and ease of modification. Specifically, cellulose nanocrystals (CNCs) improve compound stability through polyethylene glycol (PEG) adsorption, while sodium carboxymethyl cellulose (CMCNa) optimizes the liposomal structural integrity [126,127]. For example, Tian et al. developed a TEMPO-oxidized cellulose biomimetic cell wall model to protect fucoxanthin-loaded liposomes (FX-Lip), achieving a composite system with a diameter of 239.33 nm, encapsulation efficiency of 98.10%, and drug-loading capacity of 14.72%. The reduced absolute zeta potential value indicates that TCNFs (TEMPO-oxidized cellulose nanofibers) prevent coalescence by enhancing inter-droplet electrostatic repulsion. At 60 °C, fucoxanthin retention exceeded 90%, confirming effective leakage suppression by the lipid bilayer. After 15-day storage, the composite retained 97.85% stability. TCNF@FX-Lip exhibits pH-responsive characteristics: under acidic conditions, protonation of TCNF carboxylate groups forms carboxylic acids, promoting ester bond formation and reducing electrostatic repulsion-induced aggregation; in alkaline environments, carboxyl group dissociation enhances hydrophilicity and electrostatic repulsion, thereby improving stability [109].

## 4. Safety and Bioavailability of Fucoxanthin in the Novel Controlled-Release Delivery System

Before applying encapsulated fucoxanthin in food and nutraceutical industries, a comprehensive evaluation of the delivery system’s impact on its safety and bioavailability is essential. The research by Robles-García et al. demonstrates that fucoxanthin encapsulated with soybean phosphatidylcholine significantly enhances erythrocyte protective capacity by inhibiting hemolysis, photohemolysis, and thermally induced hemolysis. The encapsulation system optimizes bioavailability through increased aqueous solubility, modulated diffusion kinetics, and improved cellular uptake [107].

The shell structure predominantly comprises hydrophilic group-containing biocompatible materials (e.g., hydroxypropyl-β-cyclodextrin, zein). Given the compositional and pH variations in digestive fluids across gastrointestinal segments, encapsulation systems must exhibit dual pH/enzyme responsiveness to achieve site-controlled release. In vitro digestion experiments reveal that fucoxanthin-loaded oxidized paramylon starch achieves only 23.41% cumulative release in gastric conditions due to its dense hydrophilic shell, whereas alkaline intestinal conditions induce carboxyl group ionization, elevating cumulative release to 95.58% [99]. Palmitic acid (PA)-loaded fucoxanthin enhances intestinal absorption by upregulating SR-B1 and ABCA1 lipid transporter expression. The lipid bilayer architecture of extracellular vesicles confers superior hemocompatibility and physiological barrier penetration [105]. These vesicles primarily undergo clathrin/caveolae-mediated endocytosis in intestinal uptake and exhibit prolonged gastrointestinal retention. This encapsulation innovation holds significant value for advancing fucoxanthin safety and bioavailability [128].

## 5. Conclusions

Fucoxanthin has been the subject of considerable research interest. This phenomenon can be attributed to the multifaceted bioactive properties of the marine active substance. These properties have been demonstrated to be beneficial to human health. However, intrinsic structural vulnerabilities including thermosensitivity, photosensitivity, pH sensitivity, low aqueous solubility, and poor bioavailability significantly restrict its applications in food and pharmaceutical industries. At present, research efforts are focused on the development of delivery systems for fucoxanthin encapsulation, with the aim of increasing its bioavailability. Whilst conventional design strategies predominantly utilize natural or synthetic organic materials—such as proteins, lipids, and polysaccharides—to stabilize active compounds, regulate release kinetics, and enhance bioavailability through molecular interactions, there has been a paradigm shift towards biomimetic approaches in this field. Although comparative analyses of controlled-release systems demonstrate that polymeric nanoparticles achieve reduced particle diameters and superior encapsulation efficiency (exceeding 98%) alongside enhanced storage stability and optimized release kinetics compared to emulsion systems, albeit with more complex manufacturing processes, the field of biomimetics has emerged as a truly transformative avenue for developing next-generation delivery platforms. Among these innovations, extracellular components have garnered significant attention, with a particular focus on extracellular vesicles (EVs) due to their innate biological compatibility and targeting precision. These biomimetic systems offer a number of advantages over conventional organic systems, including low immunogenicity and significant physiological barrier penetrability. Consequently, they have the potential to enhance both biosafety and payload bioavailability to a substantial degree. Furthermore, the employment of engineered EV production strategies has the potential to circumvent the yield limitations typically associated with natural EVs, thereby facilitating the expansion of fucoxanthin’s applications across the food and pharmaceutical sectors. It is recommended that future investigations place a priority on intelligent encapsulation technologies, such as multifunctional delivery systems. Scaling up production and clinical translation continue to represent significant challenges, necessitating process optimization and cost reduction. Interdisciplinary collaborations integrating materials science, biology, and medical sciences will create novel opportunities for broadening fucoxanthin applications.

## Figures and Tables

**Figure 1 pharmaceutics-17-00889-f001:**
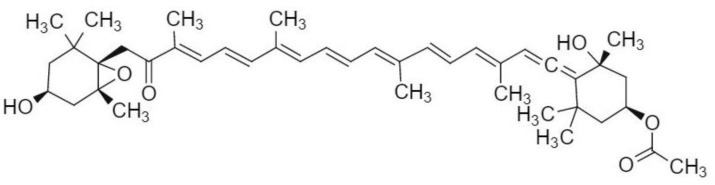
Chemical structure of fucoxanthin.

**Figure 2 pharmaceutics-17-00889-f002:**
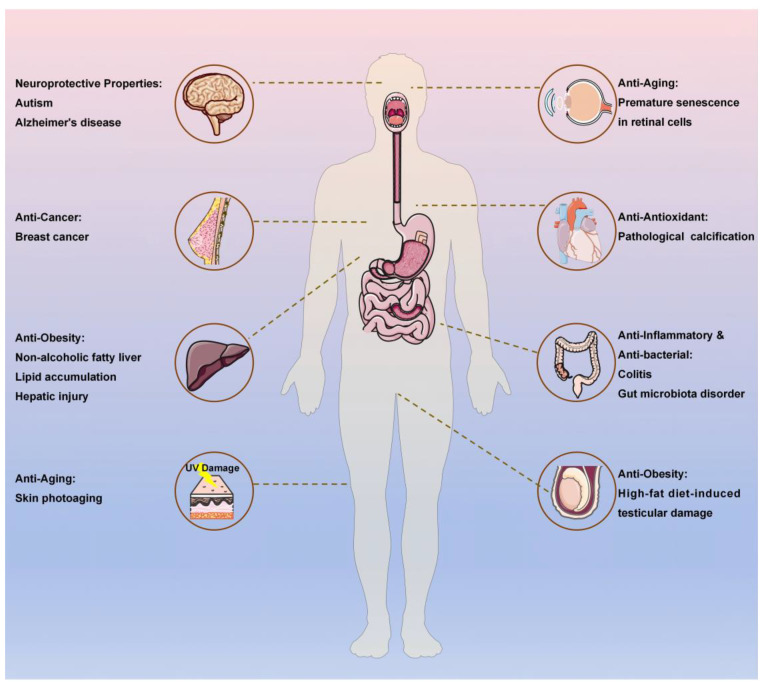
The multi-target bioactivities of fucoxanthin.

**Figure 3 pharmaceutics-17-00889-f003:**
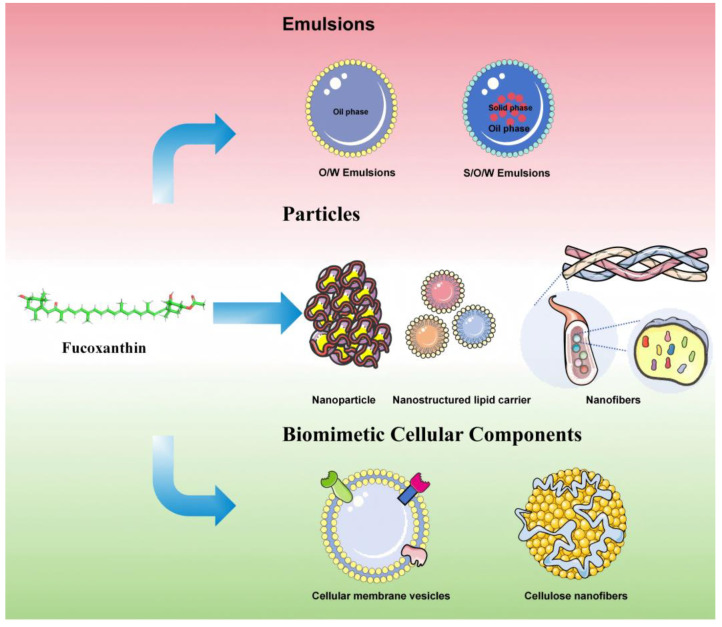
The various controlled-release systems.

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
