# Peer review of "Innovative Controlled-Release Systems for Fucoxanthin: Research Progress and Applications"

_pharmaceutics, 2025, doi:10.3390/pharmaceutics17070889_

Round 1

Reviewer 1 Report

Comments and Suggestions for Authors

This review summarizes research findings on the development of controlled-release systems using fucoxanthin. Due to the instability of fucoxanthin, encapsulation is almost essential for its various applications. Therefore, reviewing the encapsulation methods reported so far is considered meaningful.

This review begins with an introduction to fucoxanthin, followed by an overview of its biological activities, and then introduces representative encapsulation techniques along with examples of their applications. The content is well-organized overall, and in particular, the summary in Table 1 of Chapter 5 is expected to be highly useful for readers interested in this field, as it allows for a quick understanding of the characteristics of each method.

However, since the number of columns in Table 1 is quite large, there may be difficulties in layout and formatting. It is recommended to either split the table into two parts or arrange it vertically in a longer format.

Reviewer 2 Report

Comments and Suggestions for Authors

Overall a nice review of potential delivery mechanisms/systems to preserve the activity of fucoxanthin, a challenge for this type of MNP.  Below are a few suggestions to help the reader with this type of manuscript.

1) In section 2.1 line 84, since the clinical study was not done with purified fucoxanthin then I would suggest a wording change to "Clinical validation is implied by meta-analysis demonstrating that fucoxanthin-enriched seaweed supplementation...."

2) Section 2.2 first line (89) also needs to be qualified, suggest change to "The potential ani-aging efficacy of fucoxanthin...."

3) In section 2.2 line 102, what is implied by "...in a synergistic manner..."; synergistic with what and the sentence implies administration in a synergistic manner - this does not make sense pharmacologically as synergy implies with another pharmacological or physiological process.  Please clarify.

4) Section 2.3 also needs to start with a qualifying statement: "Preclinically, fucoxanthin demonstrates...."

5) Secion 2.3 line 118 suggest adding where this data is generated "... against Alzheimer's disease pathology in animal models involves suppression of..."

4) In this review article there is one large table with all the pertinent studies listed, I would suggest breaking this table into 3 associated with section 3 - Biological controlled-release systems, 4 - Safety and Biovailability and 5 - Biological activity in disease models.  This will help the reader associate the appropriate studies with the amount of data supporting activity.

5) In Section 5, if data exists it would be nice to have a comparison of pure fucoxanthin with and without the delivery system to contrast really how it improves data.

Reviewer 3 Report

Comments and Suggestions for Authors

Review of the article « Innovative Controlled-Release Systems for Fucoxanthin: Research Progress and Applications» by the authors Shiyan Wang , Mengran Guo, Zhaohui Jin 

The article focuses on fucoxanthin, a carotenoid predominantly derived from brown seaweed. It discusses the properties of fucoxanthin that are beneficial for biomedical applications, as well as various systems designed for its targeted delivery. This article will primarily appeal to researchers involved in the development of drug delivery systems. The collected material is of scientific significance; however, the main critiques pertain to the overall structure of the article.

  1. The second part of the article is dedicated to the biological activity of fucoxanthin. However, it creates an impression that the introduction does not accurately reflect the content that follows. In the introduction, it is stated that the compound may be applicable to a wide range of diagnoses, including diabetes, obesity, cancer, inflammatory diseases, cardiovascular conditions, and neurodegenerative disorders. Yet, in the second part, only the anti-obesity effects, anti-aging properties, and neuroprotective benefits are discussed. It is not until we reach part 5 that the remaining properties are addressed, as they pertain specifically to the characteristics of the encapsulated drug. I recommend integrating these sections by organizing the substances according to their properties, beginning with a discussion of the properties of the compound and concluding with its targeted delivery systems.
  2. In the introduction, the authors emphasize that controlled release systems not only improve the delivery and release of fucoxanthin at the appropriate time and location but also address several challenges associated with the compound, including its poor chemical stability, which makes it highly susceptible to photodegradation, thermal decomposition, and oxidative stress. Table 1 includes a column highlighting these capabilities of the system.However, I would suggest discussing this issue in the conclusion: reviewing and comparing the advantages and disadvantages of controlled release systems of different formulations.
  3. Another point of concern regarding the article is the absence of diagrams and figures. A diagram would be particularly beneficial in the sections discussing the properties of the compound (parts 2 and 5) and in section 3.1.

Misprints and Minor

014 "we ... summarized ... and highlights" - "... and highlighted"

021      "systems ... have gained ..., regarded as ..." - "..., which is regarded as ..."

026 "insights will provide ..., and aim ..." - no comma needed: "will provide ... and will aim ..."

068 "The Biological Activity of FUCOXANTHIN" - no uppercase needed: "... of Fucoxanthin"

081 "Hydroxy-3-methyl-glutaryl coenzyme A" - no uppercase needed; "3" lost: "3-hydroxy-3-methyl-glutaryl coenzyme A"

163 "heating at 60oC. [71]." - extra period "heating at 60oC [71]."

176 "modulation[74]." - space missing "modulation [74]."

Formatting

091 "Drosophila melanogaster" - bio object names should be in italics

106 "in vivo"/"in vitro" - multiple occurences; should be in italics

306 "fucoidan-N. sativa oil" - spaces missing; bio object names should be in italics "fucoidan - N. sativa oil"

Reviewer 4 Report

Comments and Suggestions for Authors

The review manuscript by Shiyan Wang and co-authors concerns the analysis of recent publications on the medical application of marine algae-derived carotenoid fucoxanthin. The authors focus on the biological activities of fucoxanthin, including its anti-obesity, anti-aging and neuroprotective properties. Further, the review examines a number of different nanocarriers for fucoxanthin delivery that enhance the stability and biocompatibility of this molecule. The authors pay special attention to biomimetic systems, such as extracellular vesicles and plant cell wall-related nanocarriers, as the most promising fucoxanthin delivery vehicles in terms of efficiency and safety. The recently published results on fucoxanthin encapsulation systems with controlled release are summarized in informative table 1.

The text of the review is well written and logically structured. The authors of the manuscript demonstrated a deep knowledge of issues under consideration , the list of references fully corresponds to the current level of knowledge on the fucoxanthin biomedical application and its delivery vehicles development.

I have two minor comments on this manuscript.

  1. I think, it would be better to expand the title of Section 3 as “Novel Controlled-Release Systems for Fucoxanthin Delivery”, for more clarity.
  2. Lines 152-154: “…the following classifications are made: conventional emulsions (200 nm – 100 μm), nanoemulsions (100–500 nm), and microemulsions (<100 nm) [69]”. I would like to ask the esteemed authors to check the correctness of this sentence. Usually, the word “microemulsions” semantically refers to micron-sized emulsions, and they should not be smaller than nanoemulsions. In addition, I found that the referenced article 69 does not contain at all the aforementioned classification of emulsions and cannot be cited in this context. Please check and correct this point.

Reviewer 5 Report

Comments and Suggestions for Authors

This review examines fucoxanthin, a marine carotenoid derived from algae, which encounters challenges such as instability, poor solubility, and limited bioavailability. The manuscript presents various controlled-release nanotechnologies that enhance fucoxanthin formulations. It explores the bioactivities and delivery systems utilizing organic materials for emulsions, nanoparticles, and carriers. Bioinspired systems, including extracellular vesicles, are highlighted for their improved encapsulation capabilities through immunocompatibility. The authors discuss delivery systems aimed at enhancing biosafety and therapeutic performance, thereby guiding the development of platforms for drugs and supplements.

While the review is well-structured, it is notably brief, comprising just over 10 pages, with three and a half pages dedicated to a table. There is a pressing need to expand the work by incorporating new aspects and elaborating on the existing content in greater detail.

Round 2

Reviewer 5 Report

Comments and Suggestions for Authors

Dear authors,

I extend my congratulations on the significant improvements made to your manuscript.

It is now well-prepared, and I believe it is suitable for publication in its current form.